# "You're kind of at war with yourself as a nurse": Perspectives of inpatient nurses on treating people who present with a comorbid opioid use disorder

Gabrielle Horner[1]*, Jeff Daddona[1], Deirdre J. Burke[2], Judith Cullinane[3], Margie Skeer[1], Alysse G. Wurcel[1,2]

**1** Tufts University School of Medicine, Department of Public Health and Community Medicine, Boston, Massachusetts, United States of America, **2** Tufts Medical Center, Department of Medicine, Division of Geographic Medicine and Infectious Diseases, Boston, Massachusetts, United States of America, **3** Tufts Medical Center, Department of Nursing, Boston, Massachusetts, United States of America

\* Gabrielle.horner@Tufts.edu

**Data Availability Statement:** All relevant data are within the manuscript and its supporting information files.

## Abstract

### Background

In the midst of an opioid epidemic, health care workers are encountering an increasing number of patients who have opioid use disorder in addition to complex social, behavioral and medical issues. Of all the clinicians in the hospital, nurses spend the most time with hospitalized patients who have opioid use disorder, yet there has been little research exploring their experiences in caring for this population. The objective of this study was to assess the attitudes, perceptions, and training needs of nurses in the inpatient setting when caring for patients who have opioid use disorder.

### Methods

One-on-one in-depth interviews were conducted with nurses working at a large academic medical center in Boston, MA, using a semi-structured interview guide. Nurses were recruited via email notifications and subsequent snowball sampling. Interviews were recorded, transcribed and analyzed using a grounded theory approach.

### Results

Data from in-depth interviews with 22 nurses were grouped into six themes: (1) stigma, (2) assessing & treating pain, (3) feelings of burn out, (4) communication between providers, (5) safety & security, and (6) opportunities for change. These themes were organized within four ecological levels of the Socio-Ecological Model: I) societal context, II) hospital environment, III) interpersonal interactions, and IV) individual factors. Nurses were cognizant of the struggles that patients who have opioid use disorder confront during hospitalization such as pain, withdrawal and stigma, and elaborated on how these challenges translate to professional and emotional strain among nurses. Nurses offered recommendations by which the

**Funding:** This work was supported by the National Institutes of Health (NIH), Grant number: 1KL2TR002545-01 (https://grants.nih.gov/grants/funding/ac_search_results.htm?text_curr=KL2&Search_Type=Activity) to AW. The funder had no role in study design, data collection and analysis, design to publish, or preparation of the manuscript.

**Competing interests:** The authors have declared that no competing interests exist.

hospital could streamline care for this population, including expanded role support for nurses and more structured policies regarding care for patients who present with a comorbid opioid use disorder.

## Conclusion

Our results highlight the need for the development of programs targeting both organizational culture and the inpatient nurse quality of life to ultimately enhance quality of care for patients who present with opioid use disorder.

## Background

The U.S. is in the midst of an escalating opioid crisis, with an estimated 11 million people aged 12 or older reporting misuse of prescription pain relievers in the past year [1]. In Massachusetts—a state hit hard by the opioid epidemic—it is estimated that about 5% of people meet the Diagnostic and Statistical Manual of Mental Disorders criteria for opioid use disorder (OUD), or "problematic pattern of opioid use leading to clinically significant impairment or distress."[2,3] The increasing national prevalence of OUD, as well as associated medical complications (e.g. overdose, infections, trauma), has resulted in increasing numbers of people who have OUD who are being hospitalized [4–6].

The inpatient healthcare system was ill-prepared for the rapid influx of patients who have OUD, who often present with medical complexities, co-morbid psychiatric disease, and have an increased risk of experiencing poverty, homelessness and other socio-economic barriers to health. People who have a substance use disorder (SUD) encounter stigma in the medical system reflective of a broader culture that has historically criminalized drug use and framed addiction not as a disease, but as a choice [7,8]. The direct consequences of stigma include: delay of medical care, nondisclosure of risky behaviors, rushed visits, downplaying pain, avoidance of harm reduction services (such as needle exchange programs) and decreased drug treatment completion [9–11]. One systematic review found that healthcare workers had pervasive negative attitudes towards patients with SUD, along with perceptions of violence, manipulation, and poor motivation [12]. Stigmatizing attitudes and actions of healthcare workers towards patients who have a SUD, including OUD, are inexorably linked to worse patient outcomes [12–15].

There is a paucity of research focusing on the experiences of nurses—clinicians on the front lines of caring for hospitalized people with comorbid OUD. Examination of nursing education in the context of an opioid crisis has found inadequate content related to SUD and subsequent high degrees of discomfort among nursing students in caring for patients who have OUD in the inpatient setting [16,17]. In order to improve patient outcomes, reduce stigma, and avoid clinician burn-out, the perspectives of nurses must be heard and addressed proactively. The objective of this study was to use qualitative methods to understand attitudes, perceptions and training needs of nurses working in a tertiary care hospital in Boston, MA, focusing on their experiences caring for patients who present with disordered opioid use.

## Methods

Tufts University Social, Behavioral & Educational Research IRB approved this study (#1707004 & #1707003).

## Study design

The first round of interviews was included as a part of a quality improvement project that began in April 2017, led by medical faculty and graduate students. In September 2017, concurrent IRB reviews were completed (Tufts University Social, Behavioral & Educational Research IRB) so that researchers could retrospectively use the collected data–which had originally been exempted—and continue conducting interviews.

## Participants and setting

This study took place at Tufts Medical Center, a large, urban academic medical center in Boston, MA. Nursing leadership was informed of the study and study materials approved by the Executive Director of the Center of Excellence for Nursing Research and Innovations (JC) at the hospital. Clinical nursing managers on inpatient medical units were asked to assist in recruitment of participants through sending e-mails publicizing the study. Subsequently, snowball sampling and word-of-mouth were used to recruit nurses working on non-medical floors.

The interviews and research were introduced as aiming to assess, "attitudes, perceptions and challenges nurses may encounter when treating patients with opioid addiction." We did not use the term "opioid use disorder" in the interview guide as it is not commonly accepted terminology in the nursing realm yet, but if nurses discussed people with "addiction" then we interpreted this as "OUD." All interviews were carried out using a semi-structured guide (Appendix 1) administered over 15 to 30 minutes during the work-day in private spaces in the hospital (private rooms were not always available). Participants were given the opportunity to be interviewed via phone after work hours for convenience. Interviews were audio-recorded for transcription purposes and recordings de-identified. Informed consent was verbally obtained from all participants before interviews began, ensuring anonymity and participants were given the opportunity to skip questions or end the interview at any time (Appendix 1). Interviews continued until data saturation was reached.

## Data analysis

All de-identified audio-recordings were transcribed verbatim and uploaded onto NVivo 12 Plus software. Three iterative phases of data analysis proceeded according to Birks and Mills' "traditional model" of grounded theory [18,19]. First, a preliminary codebook was developed from the first round of interviews using an open coding technique, in which broad labels were identified and as many quotes as possible categorized. This round of coding included both inductive and deductive reasoning, as the interview guide was used as a reference for code generation. Following the second round of interviews, the codebook was expanded, refined, and codes were linked via comparative analysis. The purpose of this intermediate step was to establish themes and confirm our codebook. At this stage, inter-rater reliability was ensured through comparison of one transcript independently coded by three study staff and a kappa score of 65% calculated.

The third and final step of data analysis was employment of theoretical coding. During this process, it became clear that the Socio-ecological model (SEM) was an appropriate theoretical framework to organize our themes, as it highlights the interconnectedness of systems processes with individual attitudes and perceptions [20]. We did not apply a predetermined framework to our data analysis until this final stage of theory generation, to avoid preconceptions and bias in the initial readings of transcripts. Rather, the SEM emerged as a model in which our qualitative findings could best be interpreted (Fig 1).

| Socio-Ecological Level | Theme | Opportunities |
|---|---|---|
| **I. Societal Context** | **Stigma**<br>"...you kind of have that perception that all of these patients are kind of here just for the opioids, but you definitely can't be just like labeling. Like some people do actually want to get better. You can't just label everyone as a narcotic seeking patient." (Male, 25-34) | • Streamlined pathways for discharge |
| **II. Hospital Environment** | **Safety & security**<br>"When someone has cardiac chest pain, I have a set 'this is what I do in that situation.' With this, not really. It really depends on the situation. Is the patient combative? Are they conscious? Is their family member there? Are there dirty needles involved? It's so variable that I don't think there is a set protocol saying what to do in that situation. How do I stay safe?" (Female, 25-34) | • Standardization of protocols |
| **III. Interpersonal Level** | **Assessing & treating pain**<br>"I think it is more of a patient perception that they always want to be pain free. Zero pain, zero is the goal. Whereas, you just had a major surgery, you're going to have some pain. It's just getting it to a degree that is manageable, not where you have none." (Female, 18-24)<br>**Communication**<br>"As long as the different nurses are passing along info or saying, you know, 'I'm afraid that he's going to tell you one thing and telling me something else or telling the doctor something else.' A lot of the times the nurses will pull in the doctors and say, 'Look, I feel like he's saying this to me, this to you, and this to her. Let's all go in together.'" (Female, 45-54) | • Addiction consult team & involvement of social service providers<br><br>• Team-based huddles<br><br>• Increase transparency with care plan upon admission (e.g. pain "contracts") |
| **IV. Individual Factors** | **Feelings of burnout**<br>"On the medical floor we had so many patients who were just, exhibiting drug seeking behaviors, you just get fed up with it, you know? If they really are in pain sometimes... I might not quite see it. I see it a lot of times like they're just seeking the drugs." (Male, 25-34) | • Emotional & role support for nurses<br>• Education for staff |

**Fig 1. Themes from qualitative interviews organized within the Socio-Ecologic framework with supporting quotes.**

## Results

### Participant characteristics

Twenty-two interviews were completed, with a 100% completion rate of nurses asked or who volunteered to be interviewed. All interviews were in-person except one via phone. Seventy-three percent of participants were female (n = 16) (Table 1). The mean number of years working at the study institution were 7.6 (SD = 8.9) and the mean number of years in the field of nursing were 10.5 (SD = 10.2).

### Thematic analysis

The following six themes emerged in our analysis: (1) stigma, (2) safety & security, (3) assessing & treating pain, (4) communication between providers, (5) feelings of burnout and (6) opportunities for change. The themes were situated within the SEM framework, including I) societal context, (II) hospital environment, (III) interpersonal interactions, (IV) individual factors (Fig 1). Although themes have been assigned specific levels of the SEM framework, inherent in the SEM is the bi-directionality of 'ecological levels' (e.g. stigma, on societal and individual levels) and thus intentional overlap exists [20,21].

**(1) Stigma.** Stigma was classified as a societal construct that transcends all aspects of the workplace and staff-patient interactions. Nurses frequently referred to stigma that patients who have OUD experience in the hospital and the negative impact that stigma has on healthcare delivery. They described patients being admitted with "their defenses up" or "with a kind of barrier, with a wall up." One nurse explained that patients set up a "cycle of problems,"

Table 1. Demographic characteristics of interview respondents.

| Participants | Interviewees (n = 22) |
|---|---|
| | N (%) |
| **Gender** | |
| Female | 16 (72%) |
| Male | 6 (27%) |
| **Service Line** | |
| Internal Medicine | 10 (46%) |
| Pediatrics | 3 (14%) |
| Surgery | 5 (23%) |
| Other* | 4 (18%) |
| **Age** | |
| 18–24 | 3 (14%) |
| 25–34 | 11 (50%) |
| 35–44 | 3 (14%) |
| 45–54 | 3 (14%) |
| 55–64 | 2 (9%) |
| 65–74 | 0 |
| **Years in Nursing** | |
| 5 or less | 9 (41%) |
| 6–10 | 7 (32%) |
| 11–15 | 2 (9%) |
| 16–20 | 1 (5%) |
| 21–25 | 0 |
| 26–30 | 1 (5%) |
| More than 30 | 2 (9%) |

*Includes ICU nurses & floating nurses

where "the staff perceives them to be annoying or obnoxious, then the patient can feel that. . ., [then] they're going to be a little bit meaner or less kind to the nurses in return. And it just kind of keeps going." (Female, 18–24) Another nurse reported that concerns for stigma were warranted, since "staff attitudes are obvious, you can't really hide them that well." A participant pointed out that this stigma may be rooted in lack of education, as nurses may not recognize the physical manifestations of withdrawal and cravings:

> "I think, maybe feeling judged a little bit, I don't know if by medical professionals per se but maybe people who aren't educated in opiates . . . A lot of times we don't understand the psych component or what not. Patients will get agitated and some staff members [say] 'you need to calm down' but they can't calm down, they're looking for- they're drug seeking because they want to treat their withdrawal symptoms." (Female, 45–54)

**(2) Safety and security.** Nurses described reliance on security in deescalating situations in which patients, or their visitors, exhibited aggressive behavior. A nurse explained, "we try to just keep our staff safe, but our anxiety rises as well," when describing the process of admitting a patient with a known drug addiction. Nurses discussed calling on security to check bags when patients are admitted or when there is suspicion that patient visitors are supplying non-prescribed drugs to patients. One nurse explained:

"The stories would blow your mind. . . . I call security more than most people because I've seen the worst and worst and I think people sometimes tend to give them too much leverage. I've seen it go from zero to that [snaps fingers] because their desperation is like none other . . . They will be here and know how sick they are, but if they are not being adequately medicated with narcotics, they will rip IVs out, rip dressings off, anything, just to go out and use. Real desperation you can't even describe." (Female, 55–64)

Personal safety was a concern brought up by female nurses, but not male nurses. One female nurse said, "we [nurses] protect ourselves. . .but if it's like a young man that's big, I just call public safety and have them deal with it. . .cause usually it's a male [patient]. Sometimes as a female, if it's a male patient they don't listen to them as much. Females they like can get up in your face." (Female, 25–34) Another nurse similarly stated, "more of the male nurses they are like taller and a little bit more muscular, so if it is an aggressive patient they are probably a little less likely to think they can bully that nurse around so, [the male nurses] probably don't see as much aggression." (Female, 18–24)

**(3) Assessing & treating pain.** Many nurses described an internal conflict over medicating pain, worrying that giving pain medicine would contribute to patients' addiction. A nurse captured this as: "you don't want to fuel their addiction, you don't want to set them back, but you want to treat them. So, there's just this clash of really how to go about things." (Female, 18–24) This uncertainty in when or how to treat pain in the context of addiction was complicated by nurses' professional ethos to provide relief to patients who are suffering. Another interviewee summarized this by saying, "you are kind of at war with yourself as a nurse, being like 'am I just going to medicate you because you're a drug addict and you're looking for it?' but you can't really withhold meds." (Female, 25–34)

Another source of tension involved "believing the pain," as the origin and intensity of opioid cravings may not represent pain, but addiction. Several nurses described attempts to reframe addiction as a disease rather than a personal choice in order to approach patient concerns as genuine and warranting a response. A nurse explained, "there's always debates around is it a choice or is it a disease. You know, I think the best way to approach it as a caregiver is to not get involved in that conversation at all and to really understand that this is a patient, and they need help, and we're here to help them and get them through it, you know?" (Male, 35–44) A few nurses discussed how personal experiences had reinforced more compassionate views regarding both addiction and pain. Two nurses reflected:

"Well, addiction runs in my family, so I've had some outside education just through dealing with family members. . . I think it definitely helps being educated on it, because I find I have more compassion. And treat it as a disease, and not as they're trying to be difficult or. . . it's a real problem and it's a medical issue now. I try not to be judgmental on anyone. And then when you see their families feeling so helpless too, that just helps you be more compassionate. That they've been struggling with this, and how can we help them." (Female, 45–54)

"I had my mother who was really sick growing up on, with cancer, and was on some pain meds. I saw her. . .go through some stuff, and I really felt that people, people need to be medicated and be comfortable. It's the only thing that's fair, and even if they caused their problem themselves, you know, if they brought it on themselves, or if they used drugs and that's why they're sick, they still deserve to not be in pain." (Female, 45–54)

Finally, several nurses expressed concern that pain in people with OUD is neglected because prescribers resort to less powerful narcotics that is "not going to touch them. . .[but] only aggravate them." One nurse recalled:

"I had a patient once who was in so much pain, he had a history of opioid use and we were only giving him Tylenol, Tylenol, Tylenol and we finally did some scans and he had [metastases] everywhere. So, then we were like, 'oh s**t we were only giving him Tylenol'. For me especially, that kind of put things in perspective. This guy who has been clean and we weren't treating him adequately." (Female, 45–54)

**(4) Communication between providers.** Most nurses were satisfied with provider communication and expressed comfort consulting with the care team about a patient's opioid use or suspicion of inpatient non-prescribed drug use. The value in "getting on the same page" was elaborated on by one interviewee, who stated, "usually it's the nurses who say, 'come, let's go in and talk to somebody together to make sure that we're all together and hear the same things.'" (Female, 45–54) Nurses also provided insight into how communication could be enhanced between shifts to confirm patient care goals and protocols. One nurse described the challenge of overnight shiftwork:

"Sometimes overnight. . .if the patient was asking for medications and kind of demanding, they might just give them a one-time order to calm them down. You know try to solve the immediate problem but not the grand scheme of things, because a lot of times once they get a one-time order, you know, the very next day or the next night 'well they gave it to me last night, why can't I get it now?' and it starts all over again." (Male, 45–54)

Nurses identified "staff-splitting" as a major consequence of inconsistent communication. Interviewees defined "staff-splitting" as instances in which patients use one nurse's words or actions against them to vie for increased access to pain medications. One nurse described how "patients. . .tend to take advantage of one nurse's words, like if you say the wrong thing or the wrong hours of when you're due and how much you can get for this dose. . . they kind of use that against the next nurse and like 'this guy gave me this for this pain, why aren't you giving me this? Continuity definitely is a problem because. . .for the nurse who is continuously taking care of them, it might be taxing on them mentally. . .like if they're verbally aggressive and stuff like that.'" (Male, 25–34) Nurses expressed how communication regarding pain levels and expectations should be addressed as soon as the patient is admitted in order to avoid deviations from the patient's medication regimen and accidentally trigger a sense of "false hope."

The implications of variable adherence to pain and addiction protocols was discussed. One nurse explained, "You will have one nurse that's like, 'I don't even care' and they don't crush meds and dissolve it in water and then you have another nurse who is like, 'we are going to crush these, dissolve it, and watch you drink it and do mouth checks.' That's how the patient can kind of manipulate everyone else." (Female, 18–24)

**(5) Feelings of burnout.** Burnout is defined as "a prolonged response to chronic emotional and interpersonal stressors and is characterized by hopelessness and apathy." [22] Feelings associated with burnout were common among nurses, with several expressing frustration and exhaustion in working with what they considered a more "demanding" patient population. Nurses explained how it was "hard not to take things personally" when patients were disruptive, inappropriate, and potentially dangerous. This feeling of disappointment stemmed from wanting to trust patients but often being let down. A nurse explained, "You want to believe these people, but sometimes they know exactly what to say and you always have that little feeling that I can't 100% trust you." (Female, 18–24) One nurse discussed how patients with an OUD can monopolize their time and how it's difficult to "[have] compassion for them if they are calling in every three hours. We have at least 4 or 5 patients in a day, so when there is that

one person who is constantly ringing in, they become an annoyance. . . . You're just like 'well, they just want their other fix, their next fix, they can wait five minutes.'" (Female, 18–24)

The exasperation conveyed by interviewees was balanced with sadness that accompanies watching young patients who have OUD cycle in and out of the hospital. A nurse recollected, "I had a patient who overdosed shortly after she left here and she signed out [against medical advice] . . .and she died two days after she left . . . and it just devastated me. I didn't even know her, I had her for 12 hours or 8 hours, and it devastated me." (Female, 55–64) Another nurse described how this sadness may lead to frustration:

"It's a challenge to treat some of those patients sometimes. Not only because of the management of the pain and the agitation and all of those, but sometimes the psychosocial aspect of knowing that they're going to go back out and do this all over again, and there's a very real possibility that we'll see them again in a couple days, couple weeks, couple months, so it's not just a burden on the patient, but it's a burden on the caregivers too, knowing what the possible outcomes are." (Male, 35–44)

The notion of offering futile care to patients who may not be willing or able to fully recover appeared in a few interviews. One participant connected this to an emerging ethical dilemma, stating:

"So say a heart surgeon, it's an IV drug abuser, and they have a really bad valve or an infected valve, and they have to do a surgery to replace it, and then they have to do it a second time, and then they continue doing their drugs, at what point do you stop offering life-saving surgeries? Because they keep doing these drugs that are killing them, pretty much. So that's another aspect of it, at what point do you stop offering it? You know, there are some surgeons that say, 'I'll do it twice, but I won't do it a third time'." (Male, 35–44)

**(6) Opportunities. Transitions from the hospital to the community:** At the societal level, nurses highlighted the importance of having safe and appropriate places to send patients with OUD after discharge. A nurse pointed out "a lot of times they also have social issues or family issues which makes discharging difficult and certain people who have a history of use or dependence they can only go to certain rehabs so that's another challenge too." (Female, 18–24) One nurse optimistically stated, "I think there's definitely a concerted effort not only by this hospital, but by all of the hospitals within the state of Massachusetts and all of the healthcare entities to focus on prevention and community treatment, I think that is a really good way to go." (Male, 35–44)

**Standardizing care:** Nurses mentioned how standardized protocols could facilitate limit-setting and pain expectations between the care team and the patient. Involving the whole team in establishing patients' pain goals was highly valued by interviewees. One nurse explained:

"I do find what helps is that there is firm limit-setting at the very beginning with what patients will be able to have and that is communicated throughout the entire team . . .I really like when it's me, the physician, maybe a social worker, and we're all in the room and we all hear that we are not giving you [name of narcotic] for whatever reason, and that patient is aware." (Female, 25–34)

Several nurses referred to "contracts" as an opportunity to standardize treatment and clarify expectations, however there was uncertainty as to whether contracts were used consistently:

"Supposedly, preoperatively, the doctor is supposed to let the patient know what they are going to be taking, after surgery. I obviously can't speak for how true that is but I think that if there was some sort of a contract that the patient had to sign or like an agreement on a pain regimen after surgery, just so they know like what to expect." (Female, 18–24)

A couple of nurses complimented the recently enacted "pain contract" that had been established on their floor. This "pain contract" between providers and patients aims to establish expectations regarding pain control and patient safety over the duration of the hospital stay. Nurses described the contract as being a way to avoid constantly telling patients "no" and instead being able to refer to agreed-upon terms. One nurse elaborated on the importance of language, stating, "Patients get agitated, they get the sense that we're not doing our jobs . . .it's frustrating to the patient who is continually being told 'no, no' . . .there should be keywords that may help people deescalate rather than 'no, no, the doctor said no' . . . there has to be something else." (Female, 25–34) In this sense, consistency of language could mitigate safety concerns as patients are in the loop with regard to care plans.

**Emotional support:** Interpersonal emotional and role support was identified as an important yet neglected aspect of hospital medicine for both patients and nurses themselves. One nurse commented:

"I think, the emotional side has to be taken into consideration more, not just where they're at medically. Like, it's always 'what kind of program are they in, are they planning on getting in any program', and blah blah blah. And we kind of forget about, like, they're a new mother, and where are they at emotionally with that? We're so obsessed about that one aspect of their life that we kind of miss out on a lot of the other aspects, if that makes sense." (Male, 25–34)

Another participant suggested the use of patient advocates to assist patients along with their families:

"Maybe just a liaison for them [and] more education for the staff just to kind of put them in these patients' shoes a little bit, like you don't know what these people have been through. Like no one wants to be an addict, but, I mean, they're an addict for whatever reason, and it is what it is, and you just have to get over it yourself because if you have, if you're judgmental about it, I mean, I've seen lots of staff who are not so kind to these families, and it's really not any reason for it." (Male, 25–34)

One reason underpinning the need for emotional support for nurses themselves is the prevalence of addiction among healthcare providers. One nurse summarized this important point:

"So, the other thing, so about 10% of healthcare providers become addicted themselves. We are surrounded by narcotics and have easy access to them. I'm on the nurse pharmacy committee that actually audits this and does surveillance and all that kind of stuff. So we actually had a nurse come talk to us and some of our leaders about how it happened to him, so I think there's nothing better than a story." (Male, 35–44)

**Educational needs:** Overwhelmingly, nurses expressed interest in learning more about OUD and how to improve care for this population. When asked if they would participate in an educational session on addiction that could be used as a CEU credit, interviewed nurses unanimously replied 'yes.' Nurses recommended a number of ways to structure training, offering

both content suggestions and learning methods that they perceived to be effective. Some nurses discussed how trainings should be less academic and instead focus on the realities of drug abuse, including terminology and dose information. Also, a few nurses suggested that the hospital bring in a speaker who has overcome an OUD. Similarly, a nurse suggested meetings and discussions with staff and specialists working in detoxification and rehabilitation centers, which may inform and strengthen their approach to patients who have OUD.

In response to indecision over pain management among patients with opioid addictions, nurses requested more training on if or when to start detoxing in the hospital, how to best educate patients about different treatments for pain, as well as potential drug interactions. One nurse called for a comprehensive look into best practices on treatment of addiction in the hospital setting, recognizing the inadequacies and inconsistencies of current practices:

> "I think as an institution or as a profession we need to think about how we do want to treat these patients, and do we want to have them start detoxing while they're in the institution, or do we want to keep them stable? And how to address it, do we individualize it depending on what's going on? I think it's probably not addressed consistently enough on how to treat it, and it's just treated as a secondary problem." (Male, 35–44)

## Discussion

As the healthcare system struggles to accommodate an increasing number of people with comorbid OUD, an inter-professional "all hands-on deck" approach is needed [23]. In this study, we found that in interactions with hospitalized patients who have an OUD, nurses experienced challenges related to managing patients' pain, overall communication, and threats to personal safety which collectively contributed to feelings of burnout. Insufficient and outdated training magnified these challenges, while stigma in many cases thwarted the therapeutic relationship between nurses and patients.

Strategies to augment nurses' role and motivation to improve care for patients who have OUD in the inpatient setting must be multifactorial, starting with education of nurses and supported by organizational change. Our findings indicate that nurses are willing to learn and develop these skills but lack a clear purpose or direction (i.e. role adequacy and role legitimacy). One nurse drew the comparison: "When someone has cardiac chest pain, I have a set 'this is what I do in that situation.' With this, not really." (Female 25–34) Measures of role adequacy (feeling knowledgeable about one's work) and role legitimacy (feeling of having the right to inquire and act upon certain issues) have been shown to affect attitudes towards people who have OUD. If nurses are not confident in their ability to assist in SUD treatment and address the biopsychosocial facets of addiction, they are less likely to actively engage with patients [24]. In our analysis, this discomfort manifested in reported distrust of patients; the term "dissonant care management" has been used to define the sense of detachment that can develop which was alleviated through "seeing the person behind the patient."[25] This echoes the internal conflict nurses in our study described with difficulty separating the person from their addiction.

Efforts must be made to re-humanize care for people who have OUD in order to mitigate both burnout and stigma. One plan of care model which emphasizes patients' 'activities of living' and independence has been described as benefiting patients in the hospital who sense a lack of autonomy [26]. This approach, called the Roper, Logan & Tierney model of nursing, aims to enhance patient self-esteem and ultimately outcomes through shared decision-making. Giving nurses the tools to not just learn about addiction but also empower those struggling with addiction can have dual provider-patient benefits. There is a consistent positive correlation between healthcare workers' attitudes towards patients with disordered drug use and their

degree of familiarity with substance use problems, increased contact with this group, and more confidence in treatment [27,28].

There have been calls to shine a spotlight on organizational culture as opposed to solely relying on additional training when designing interventions to improve staff attitudes towards working with people who have SUD [29,30]. Training on SUD management without an institutional framework for addressing these issues is not only futile, but perhaps counterproductive [31]. Although nurses in our study highlighted the importance of education in improving attitudes, one large research study found that workplace SUD education was only influential in improving nurses' therapeutic attitudes when coupled with role support [32]. A top-down approach, starting with organizational mission statements from the institution are essential, as they can facilitate a sense of purpose and motivation [30,33]. Equally important is expanded role support, such as opportunities for clinical supervision, a formal mentoring program, and a stress management program for employees [33].

In this context of advanced role support, there are evidenced-based educational topics that nurses in our study recommended. These include sessions in which individuals who have overcome addiction come in to share their stories as well as discussions led by experts in the field. An effort to deliver material that reframes addiction as a disease and highlights the social determinants of health are particularly valuable at targeting stigmatizing attitudes [34]. Another educational gap involves training on trauma-informed care, which is relevant given the high degree of trauma experienced by PWUD compared to the general population. The concept of secondary traumatic stress has been proposed as contributing to lower job satisfaction and occupational commitment among nurses exposed to higher levels of patients with trauma histories [35,36]. Again, these proposed trainings are most valuable when delivered with accompanying role support.

Nurses in our study supported organizational changes such as a more structured clinical approach (e.g. pain contracts or agreements) with early and explicit conversations about pain management. It is important to note that the concept of "contracts" with people who have OUD in the inpatient setting has been debated, especially surrounding the ethics of using contracts as a way to deny further care if they are broken [37]. Nurses in our study appreciated a contract as a tool to standardize medical care and give them more confidence in working with patients who have increasingly difficult pain and behavioral issues. Structural changes may also include daily team huddles with all members of the care team. Additionally, formalized debriefing sessions after a negative patient encounter could identify ways to avoid such situations while fostering unity among staff.

These results also revealed that nurses, especially females, encountered threatening situations involving patients who have OUD; unfortunately, this is not surprising given nurses are subjected to the highest rates of verbal and physical assaults in the workplace compared to all other health professionals because they have the greatest contact time with patients [38]. Workplace violence has been associated with increased rates of burnout, job dissatisfaction, and decreased productivity among nurses; accumulated stress in working with people who use drugs (PWUD) translates to higher rates of intention to change jobs [38,39]. Furthermore, the effects of feeling fearful of or manipulated by people who have OUD leads nurses to assume an authoritative rather than caregiving role, as they begin to "police" patients rather than continue with a patient-centered approach [40]. These hostile interactions contribute to a 'cycle of problems' and perpetuate stigma against PWUD that was described by nurses in our study. To break this cycle, safe and supportive hospital environments must be prioritized to provide the foundation on which education can be most impactful.

SUD is a medical disease, and there is increasing evidence that early addiction management during hospitalization is crucial to improving outcomes [41,42]. Embracing this concept, one

hospital implemented a comprehensive intervention to improve quality of care for people with addiction, including an inpatient addiction medicine consultation team, rapid-access pathways to post-hospital opioid abuse treatment, and a medically enhanced residential care model. To better support nurses, this intervention included Patient Safety Care Plans which laid out behavioral (both verbal and physical) expectations as well as explicit protocols for de-escalating conflict [43]. This structural change was complemented with education, for example on induction of buprenorphine. Follow-up analysis of this program revealed decreased feelings of burnout among providers as well as a shifting perspective of addiction as "a medical illness, not a moral choice".[8,44] Institution-wide interventions aimed at improving outcomes for people with opioid addictions can concurrently improve provider wellness and perceptions of safety.

Finally, national nursing organizations have put forth statements encouraging nurses to become advocates for individuals with addiction through learning and teaching about harm reduction strategies to help people with OUD avoid hospitalization [45]. For example, offering information about basic wound care, overdose prevention and intervention, phlebotomy skills and safe sexual health practices is a way for nurses to strengthen the therapeutic alliance with patients while sharing medical expertise [46]. Similarly, clinical checklists have been developed to organize key health issues among patients who inject drugs, including addiction treatment, overdose prevention, and infectious diseases prevention as a way to not neglect any aspect of patient care [47]. Nurses can also advocate for inpatient initiation of medication assisted treatment (MAT). Studies have found low rates of drug treatment plans outlined in the discharge summaries for PWUD as well as suboptimal levels of addiction medicine consultations, which are associated with increased rates of treatment completion, initiation of MAT and reduced risk of readmission [42,48]. Advocacy for both harm reduction strategies and MAT, while occurring in the hospital environment, reflect social constructs, as their widespread adoption has been debated in the political sphere despite their proven effectiveness at promoting health [49,50].

## Limitations

There are limitations to this study that should be acknowledged. First, the study took place in one academic medical center in Boston, MA, and so may not be generalizable to other parts of the country or rural areas. Results are also limited by the fact that all nurses worked on inpatient floors and most respondents were white women. The perspectives and experiences of nurses who are not white may be entirely different from nurses who are white. Further, we were unable to stratify our results by service; a meta-analysis on compassion fatigue and burnout among nurses found disparate outcomes based on hospital departments [51]. Our sample size did not permit meaningful comparison between departments, though each of our themes included data from a range of nurses. Study staff and participants were affiliated with the same institution and so when replying to interview questions, participants may have been subject to social desirability bias. However, interviewers had never met participants before and our impression was that interview responses were candid. In light of our results revealing the importance of organizational change, future studies should incorporate the viewpoints of workers further removed from patient care (e.g. hospital leadership). Understanding their perceptions of the barriers to care could better contextualize our results within an organizational framework.

## Conclusion

Nurses in a hotspot of the nation's ongoing opioid epidemic face personal and professional challenges when working with hospitalized patients who have OUD. We found that nurses

were motivated to expand the scope and quality of care for patients with comorbid OUD yet lacked the skills and support to do so. In all, an organizational culture shift paired with meaningful educational opportunities is required for nurses to harness their caregiving capacity and optimize outcomes for a frequently neglected patient population.

## Supporting information

**S1 Appendix. Interview guide with informed consent.**
(DOCX)

## Acknowledgments

We would like to thank Sarah Andebrhan, MPH, Celina Rogers, JD, MPH, and Rachel Rothstein, MPH, for their contributions to the initial quality improvement report. We would also like to thank all of the nurses and hospital staff who supported data collection.

## Author Contributions

**Conceptualization:** Gabrielle Horner, Judith Cullinane, Margie Skeer, Alysse G. Wurcel.

**Data curation:** Gabrielle Horner, Jeff Daddona, Deirdre J. Burke, Alysse G. Wurcel.

**Formal analysis:** Gabrielle Horner, Jeff Daddona, Deirdre J. Burke, Margie Skeer, Alysse G. Wurcel.

**Funding acquisition:** Alysse G. Wurcel.

**Investigation:** Gabrielle Horner, Jeff Daddona, Margie Skeer, Alysse G. Wurcel.

**Methodology:** Gabrielle Horner, Margie Skeer, Alysse G. Wurcel.

**Software:** Jeff Daddona, Deirdre J. Burke.

**Supervision:** Judith Cullinane.

**Validation:** Margie Skeer, Alysse G. Wurcel.

**Writing – original draft:** Gabrielle Horner, Alysse G. Wurcel.

**Writing – review & editing:** Gabrielle Horner, Jeff Daddona, Deirdre J. Burke, Judith Cullinane, Margie Skeer, Alysse G. Wurcel.

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
