## [Decision Letter · Decision Letter 0]

24 Jul 2019

PONE-D-19-14998

“You’re kind of at war with yourself as a nurse”: Perspectives of Inpatient Nurses on Treating People with Opioid Use Disorder

PLOS ONE

Dear Ms Horner,

Thank you for submitting your manuscript to PLOS ONE. After careful consideration, we feel that it has merit but does not fully meet PLOS ONE’s publication criteria as it currently stands. Therefore, we invite you to submit a revised version of the manuscript that addresses the points raised during the review process.

We would appreciate receiving your revised manuscript by Sep 07 2019 11:59PM. To enhance the reproducibility of your results, we recommend that if applicable you deposit your laboratory protocols in protocols.io, where a protocol can be assigned its own identifier (DOI) such that it can be cited independently in the future. For instructions see: http://journals.plos.org/plosone/s/submission-guidelines#loc-laboratory-protocols

We look forward to receiving your revised manuscript.

Kind regards,

Carla Treloar

Academic Editor

PLOS ONE

Additional Editor Comments (if provided):

Comments from two reviewers have been received. While the reviewers see potential in your paper, both have asked for some revisions. These include the literature review, and Reviewer 1 provides some specific pointers to expanding and elaborating the literature reviewed. Further, Reviewer 1 makes very salient points about the quantitative/survey section - questioning what this adds when the main result (males have more positive attitudes) rests on a very small sample size. I would urge the authors to consider the suggestions of Reviewer 1 in relation to both the qualitative and quantitative data. The qualitative data shows potential, but lacks a conceptual framework - going deeper into the literature as suggested may help to strengthen this aspect of the paper. Further, both reviewers point to the language used to describe people who use opioids and ask for revision to this.

Journal Requirements:

2) Please provide additional details regarding participant consent. In the ethics statement in the Methods and online submission information, please ensure that you have specified how verbal consent was documented and witnessed.

3) Thank you for including your ethics statement; "Tufts University Social, Behavioral & Educational Research IRB #1707004 & #1707003. Oral consent was obtained from all interview participants. Consent was not obtained from survey participants as data were analyzed anonymously and no identifying information was collected."

Please amend your current ethics statement to confirm that your named institutional review board or ethics committee specifically approved this study.

Reviewers' comments:

Reviewer's Responses to Questions

**Comments to the Author**

1. Is the manuscript technically sound, and do the data support the conclusions?

Reviewer #1: Partly

Reviewer #2: Yes

2. Has the statistical analysis been performed appropriately and rigorously? 

Reviewer #1: N/A

Reviewer #2: Yes

3. Have the authors made all data underlying the findings in their manuscript fully available?

Reviewer #1: Yes

Reviewer #2: Yes

4. Is the manuscript presented in an intelligible fashion and written in standard English?

Reviewer #1: Yes

Reviewer #2: Yes

5. Review Comments to the Author

Reviewer #1: Thank you for the opportunity to review this paper which focuses on an important topic – assessing the perspectives of nurses who work with people who use opioids. Nurses are often the front-line workers and it is therefore important to understand their feelings about working with people who use drugs and assess ways to ensure their work environment is positive.

While this study is interesting and could make a valuable contribution to the existing body of research in the area, I think it needs some major revisions to make it suitable for publication. There are two main concerns that I have, firstly with the conceptual understanding and design of the study, and secondly with the limited review of existing literature in which to situate and understand the findings. I will discuss these issues in more detail below.

1. Study design

I think this paper should only focus on the qualitative research reported in this study. This research is interesting and with a more in-depth analysis could really contribute to our understanding of nurses’ feelings and perspectives about working with people who use opioids. I feel that the quantitative data add little to the paper and seems to just be an add on to otherwise potentially interesting data. As I understand it, even the conclusions that are drawn in relation to the quant analysis – that males have more positive attitudes – is problematic as only 11 males completed the survey. Hence, I would not include this data in the paper at all and only focus on the qualitative data.

On the other hand, the qualitative data is interesting, but definitely requires a more in-depth analysis. A depth analysis may have been sacrificed at the expense of including the quant data in the paper as well. I would refocus the paper to think about how this qual data can be more meaningfully interpreted particularly in relation to existing literature which needs to be included.

So, for example stress and burnout among alcohol and drug (AOD) workers is interesting and there is existing literature which you could relate to your findings even though most of these papers are on health workers more broadly not nurses specifically.

Duraisingam, et al. (2009). The impact of work stress and job satisfaction on turnover intentions: A study of Australian specialist alcohol and other drug workers. Drugs: education, prevention and policy, 16(3), 217-223.

Skinner et al (2009). Health professionals attitudes towards AOD-related work: moving the traditional focus from education and training to organisational culture. Drugs prevention and policy. Drugs, Education Prevention and Policy, 16(3), 232-249

Roche & Nicholas (2016). Workforce development: An important paradigm shift for the alcohol and other drugs sector. Drugs: Education, Prevention and Policy, 1-12

Additionally, there is other work on AOD workers attitudes (see below) and, in particular, studies that have found nurses to have more negative implicit attitudes towards their AOD clients that medical staff (see Brener, von Hippel & Kippax, 2007). This is interesting research to reflect on in relation to your qualitative findings, as is the idea of the influence of contact with these clients or size of caseload, hours worked, type of service provided. There is a need to think more about these findings, what they may mean and how they can be understood in relation to other literature.

Lovi & Barr (2009). Stigma reported by nurses related to those experiencing drug and alcohol dependency: A phenomenological Giorgi study. Contemporary Nurse, 166 -178

Ewer et al (2015). The prevalence and correlates of secondary traumatic stress among alcohol and other drug workers in Australia. Drug and Alcohol Review, 34 (30, 252-258.

Additionally, I think it may be better to focus on some key sections and present a more in-depth analysis of this data rather than a cursory/descriptive analysis of so many themes. The themes can be grouped – ie related to workplace outcomes for staff (burnout, education needs) and client related issues (safety and security, pain management). You could also include more quotes from the tables at the end in the body of the paper and integrate them into your discussion of the results – I think those tables in the appendix are too long anyway.

There are some really interesting practical findings in the Discussion, particularly at the end of page 22 and on page 23 and if you focus on this once the qual data has been comprehensively analysed I think the paper can make an interesting contribution to supporting AOD nurses and to workforce development activities among AOD nurses.

2. More thorough review of the literature and theoretical background

As I have already noted in the discussion above I think this study could be better situated in the existing literature and a more thorough search of literature in the area is required. There is also the issue of using a conceptual or theoretical lens through which to understand this work particularly if you focus on the qualitative data. There is a workforce related theoretical literature or a social psychology literature on attitudes and social cognitions that can be used to frame your introduction. Which conceptual frame you use can depend on how you to choose to focus the paper going forward but it definitely needs some reference to a conceptual frame.

I think the paper is also missing an explanation of why this may be a difficult area to work in or why these clients may be hard to work with. Many clients who use drugs have complex mental and physical health issues while also being socially disadvantaged. This may make managing their care difficult for health workers. Additionally, drug use is highly stigmatised, particularly as it is illegal, and this stigma may influence health workers attitudes towards this group. Health workers themselves may have also experienced stigma (‘courtesy stigma’) if they work with clients who use drugs. While you don’t have to go into detail about these issues it is important to make some reference to the broader context in which drug use and drug treatment occur.

Other issues

Some other issues I noted - the methods are not described in enough detail. For example, why are nurses chosen as the target sample above other medical staff (this is probably easy to justify but it needs some justification). How were the nurses recruited, were they given the details of the researcher to contact? Where were the face- to -face interviews conducted? What was the consent process? Were interviews recorded? The key areas in the interview schedule should also be provided.

Too much focus on the limitations of the research – this looks like a list of things that are wrong with the study rather than an integrated discussion about some of the key shortcoming of the research and how these can/should be addressed in future research

By using terms such as disorder and substance abuse there is some re-stigmatising of an already highly stigmatised group – language is important in framing attitudes and I think the authors should pay some attention to this.

Overall, I really think this paper could benefit from a more in-depth analysis of the qualitative data situated in more of the related literature and framed by a clear theoretical/conceptual understanding. The quantitative data should be removed as it really adds little value, but the rest of the paper can be linked to some important applied outcomes and hence with major revisions can make a valuable contribution to area.

Reviewer #2: The paper addresses an urgent workforce development issue common to many countries, that of frontline health professionals' willingness and capacity to provide health services to patients with a drug-related problem and/or addiction.

Strengths:

-clearly identified research gap

-mixed methods

-qual and quant methodology and analysis sound (low response rate for survey is not optimal, but not uncommon for this type of study)

-acknowledgement of study limitations is reasonable

-the discussion examines systemic and organisational approaches to support health professionals' safety, confidence and skill. This is a significant strength of the study, as much work in this area does not reach beyond recommendations for more training, which as the authors note has limited impact in the absence of systemic/organisational changes to support the preferred work practices.

Issues to consider:

1. There has been quite a lot of research on health professionals' attitudes towards AOD users. The lit review is a little thin in this regard. This may be out of necessity to meet journal guidelines on citations/word count. If there is opportunity to add another para on research in this area, it would strengthen the intro. Take the author's point that research on clients with opioid-related problems may be small, but there is a large literature on health professionals' attitudes towards drug users in general which is relevant here.

2. Terminology. In my country (Australia) we don't use terms such as 'abuse drugs', as the view is that it contains a strong negative judgement element. Instead we use term such as problematic use, or drug-related problems/issues etc. The authors may wish to consider adjusting terminology towards more neutral phrases (eg line 127).

3. Conclusion. Consider strengthening the observations regarding the need for systemic/structural/team based changes to work practices by putting a note in the conclusion to reinforce these observations.

6. PLOS authors have the option to publish the peer review history of their article (what does this mean?). If published, this will include your full peer review and any attached files.

Reviewer #1: No

Reviewer #2: Yes: Dr Natalie Skinner

---

## [Author Response · Author response to Decision Letter 0]

9 Sep 2019

Reviewer #1: Thank you for the opportunity to review this paper which focuses on an important topic – assessing the perspectives of nurses who work with people who use opioids. Nurses are often the front-line workers and it is therefore important to understand their feelings about working with people who use drugs and assess ways to ensure their work environment is positive.

While this study is interesting and could make a valuable contribution to the existing body of research in the area, I think it needs some major revisions to make it suitable for publication. There are two main concerns that I have, firstly with the conceptual understanding and design of the study, and secondly with the limited review of existing literature in which to situate and understand the findings. I will discuss these issues in more detail below.

Comment 1: I think this paper should only focus on the qualitative research reported in this study. This research is interesting and with a more in-depth analysis could really contribute to our understanding of nurses’ feelings and perspectives about working with people who use opioids. I feel that the quantitative data add little to the paper and seems to just be an add on to otherwise potentially interesting data. As I understand it, even the conclusions that are drawn in relation to the quant analysis – that males have more positive attitudes – is problematic as only 11 males completed the survey. Hence, I would not include this data in the paper at all and only focus on the qualitative data.

Response 1: Thank you for these thoughtful and thorough comments. We appreciated reading them and believe that our paper has significantly improved as a result. We have removed the quantitative data section, allowing a more in-depth exploration of qualitative themes and review of the literature. 

Comment 2: On the other hand, the qualitative data is interesting, but definitely requires a more in-depth analysis. A depth analysis may have been sacrificed at the expense of including the quant data in the paper as well. I would refocus the paper to think about how this qual data can be more meaningfully interpreted particularly in relation to existing literature which needs to be included. So, for example stress and burnout among alcohol and drug (AOD) workers is interesting and there is existing literature which you could relate to your findings even though most of these papers are on health workers more broadly not nurses specifically. Duraisingam, et al. (2009). The impact of work stress and job satisfaction on turnover intentions: A study of Australian specialist alcohol and other drug workers. Drugs: education, prevention and policy, 16(3), 217-223; Skinner et al (2009). Health professionals attitudes towards AOD-related work: moving the traditional focus from education and training to organisational culture. Drugs prevention and policy. Drugs, Education Prevention and Policy, 16(3), 232-249; Roche & Nicholas (2016). Workforce development: An important paradigm shift for the alcohol and other drugs sector. Drugs: Education, Prevention and Policy, 1-12

Response 2: In addition to incorporating these manuscripts in the discussion, we intensified our literature review to touch on the important topic of burnout among AOD workers. Further, we explored the ways in which workforce development and organizational culture can mitigate stress among nurses who care for patients who have opioid use disorder, citing several influential papers in this field. Our Discussion now revolves around the theme of workforce development as the bedrock of meaningful change in terms of nurses’ quality of life and consequently, patients’ quality of care. 

Comment 3: Additionally, there is other work on AOD workers attitudes (see below) and, in particular, studies that have found nurses to have more negative implicit attitudes towards their AOD clients than medical staff (see Brener, von Hippel & Kippax, 2007). This is interesting research to reflect on in relation to your qualitative findings, as is the idea of the influence of contact with these clients or size of caseload, hours worked, type of service provided. There is a need to think more about these findings, what they may mean and how they can be understood in relation to other literature.

Lovi & Barr (2009). Stigma reported by nurses related to those experiencing drug and alcohol dependency: A phenomenological Giorgi study. Contemporary Nurse, 166 -178

Ewer et al (2015). The prevalence and correlates of secondary traumatic stress among alcohol and other drug workers in Australia. Drug and Alcohol Review, 34 (30, 252-258.

Response 3: Our Introduction elaborates on the unique position of nurses in the care they provide and why they may be particularly susceptible to burnout. In the Introduction we also provided more detail into how negative attitudes are shaped not only by personal experience, but environmental factors and our Discussion expounds upon this important point. As for what makes nurses distinct from other medical staff, we detailed how they have the most contact time with patients and this can lead to unique challenges (for example, exposure to workplace violence). 

Comment 4: Additionally, I think it may be better to focus on some key sections and present a more in-depth analysis of this data rather than a cursory/descriptive analysis of so many themes. The themes can be grouped – i.e. related to workplace outcomes for staff (burnout, education needs) and client related issues (safety and security, pain management). You could also include more quotes from the tables at the end in the body of the paper and integrate them into your discussion of the results – I think those tables in the appendix are too long anyway.

Response 4: While we used the same 6 themes that had emerged in our initial Grounded Theory approach to data analysis, we found that these themes could be subsequently organized within the Socio-ecological model. We chose to situate our themes with the Socio-ecologic model to provide a comprehensive overview of what we heard from nurses and represent the interconnectedness of potential interventions. Our results section was then able to walk through this model to better orient the reader. The table at the end of the paper was removed and the most impactful quotes include in the text or in Figure 1. 

Comment 5: There are some really interesting practical findings in the Discussion, particularly at the end of page 22 and on page 23 and if you focus on this once the qual data has been comprehensively analysed, I think the paper can make an interesting contribution to supporting AOD nurses and to workforce development activities among AOD nurses.

Response 5: We agree, several interesting interventions have been implemented to enhance the care for patients who have OUD along with the AOD staff who care for them inpatient. When the quantitative portions of the paper were removed, we were able to elaborate on these programs in our Discussion. 

Comment 6: More thorough review of the literature and theoretical background - As I have already noted in the discussion above I think this study could be better situated in the existing literature and a more thorough search of literature in the area is required. There is also the issue of using a conceptual or theoretical lens through which to understand this work particularly if you focus on the qualitative data. There is a workforce related theoretical literature or a social psychology literature on attitudes and social cognitions that can be used to frame your introduction. Which conceptual frame you use can depend on how you to choose to focus the paper going forward but it definitely needs some reference to a conceptual frame. 

Response 6: We have broadened our review of the literature and added more details to the conceptual frame.

Comment 7: I think the paper is also missing an explanation of why this may be a difficult area to work in or why these clients may be hard to work with. Many clients who use drugs have complex mental and physical health issues while also being socially disadvantaged. This may make managing their care difficult for health workers. Additionally, drug use is highly stigmatised, particularly as it is illegal, and this stigma may influence health workers attitudes towards this group. Health workers themselves may have also experienced stigma (‘courtesy stigma’) if they work with clients who use drugs. While you don’t have to go into detail about these issues it is important to make some reference to the broader context in which drug use and drug treatment occur.

Response 7: We have deepened our discussion of stigma. In our introduction we expanded upon the topic of stigma and the implications of stigma on the health outcomes of PWUD. In our discussion, we touched on how our results elucidated ongoing stigma in the hospital setting and how stigma transcends societal, interpersonal and individual level interactions. We explored the topic of secondary traumatic stress and “courtesy” stigma that nurses may experience in working with this population. Through the lens of the Socio-ecologic model, we hoped to emphasize a broader context in which nurses work and their attitudes reflect the settings in which they work. 

Comment 8: Some other issues I noted - the methods are not described in enough detail. For example, why are nurses chosen as the target sample above other medical staff (this is probably easy to justify but it needs some justification). How were the nurses recruited, were they given the details of the researcher to contact? Where were the face- to -face interviews conducted? What was the consent process? Were interviews recorded? The key areas in the interview schedule should also be provided.

Response 8: We elaborated on why nurses were chosen as study participants (voices under-sampled overall and studies that indicate insufficient information on substance use disorders in the nursing curriculum). The interview methods were expanded upon significantly and these details provided. We also included important information into how the data was analyzed using grounded theory. 

Comment 9: Too much focus on the limitations of the research – this looks like a list of things that are wrong with the study rather than an integrated discussion about some of the key shortcoming of the research and how these can/should be addressed in future research

Response 9: The limitations section was revised after careful consideration of our results in the context of existing literature. 

Comment 10: By using terms such as disorder and substance abuse there is some re-stigmatising of an already highly stigmatised group – language is important in framing attitudes and I think the authors should pay some attention to this.

Response 10: We completely agree our paper must avoid the stigmatizing and degrading language that has historically been ascribed to people with problematic drug use. We hope to reflect the upmost respect towards this patient population in our writing; this is perhaps why we feel comfortable using the medically-appropriate terms “opioid use disorder” and “substance use disorder” in our paper, as these have been defined by the DSM V. Further, in straying from this concrete medical terminology, we deviate from the original aim of the research proposal: to enhance the care for patients at our hospital with clinical opioid use disorder. Our interview guide was purposefully created to inquire about patients who use opioids and the healthcare of this population. Finally, the citations included in our paper use the terms “SUD” and “OUD.” 

Comment 11: Overall, I really think this paper could benefit from a more in-depth analysis of the qualitative data situated in more of the related literature and framed by a clear theoretical/conceptual understanding. The quantitative data should be removed as it really adds little value, but the rest of the paper can be linked to some important applied outcomes and hence with major revisions can make a valuable contribution to area.

Response 11: We have removed the quantitative data, and we agree it makes the paper much stronger. 

Reviewer #2: The paper addresses an urgent workforce development issue common to many countries, that of frontline health professionals' willingness and capacity to provide health services to patients with a drug-related problem and/or addiction.

Strengths:

-clearly identified research gap

-mixed methods

-qual and quant methodology and analysis sound (low response rate for survey is not optimal, but not uncommon for this type of study)

-acknowledgement of study limitations is reasonable

-the discussion examines systemic and organisational approaches to support health professionals' safety, confidence and skill. This is a significant strength of the study, as much work in this area does not reach beyond recommendations for more training, which as the authors note has limited impact in the absence of systemic/organisational changes to support the preferred work practices.

Issues to consider:

Comment 1: There has been quite a lot of research on health professionals' attitudes towards AOD users. The lit review is a little thin in this regard. This may be out of necessity to meet journal guidelines on citations/word count. If there is opportunity to add another para on research in this area, it would strengthen the intro. Take the author's point that research on clients with opioid-related problems may be small, but there is a large literature on health professionals' attitudes towards drug users in general which is relevant here.

Response 1: Thank you for taking the time to review and provide thoughtful feedback on our paper. We have retained the citations of papers which investigate pervasive negative attitudes among healthcare staff towards people who use drugs (e.g. van Boekel, 2012). Further, we have elaborated on how these negative attitudes are rooted in a culture of drug criminalization and how these social constructs, most notably stigma, impact health outcomes. Our Discussion section again highlights research in this area and explains how nurses in particular are susceptible to developing negative attitudes towards people who use drugs. 

Comment 2: Terminology. In my country (Australia) we don't use terms such as 'abuse drugs', as the view is that it contains a strong negative judgement element. Instead we use term such as problematic use, or drug-related problems/issues etc. The authors may wish to consider adjusting terminology towards more neutral phrases (eg line 127).

Response 2: We appreciate your careful reading and sensitivity towards language. We recognize the importance of avoiding preconceptions and stigmatizing language in our paper. However, we decided to continue to use the terms “opioid use disorder” and “substance use disorder” as they are DSM-defined diseases and accepted terms in the literature. Further, much of the literature cited, including papers on stigma towards patients who have a SUD, use these terms and thus we did not want to deviate from the language of existing literature. 

Comment 3: Conclusion. Consider strengthening the observations regarding the need for systemic/structural/team based changes to work practices by putting a note in the conclusion to reinforce these observations.

Response 3: We agree that strategies targeting team-based, organizational change are most impactful when it comes to improving the care for people with problematic drug use. We modified our Discussion to highlight the importance of this ‘top-down’ approach to workforce development. Further, inclusion of the Socio-ecologic model highlights the overarching influence of environmental context on individual staff attitudes and perceptions towards people who use drugs. 

Thank you again for your time reviewing this article. We are excited about the potential that our article will be published in your journal. Please reach out with any questions or concerns.

---

## [Decision Letter · Decision Letter 1]

11 Oct 2019

“You’re kind of at war with yourself as a nurse”: Perspectives of Inpatient Nurses on Treating People who present with a Comorbid Opioid Use Disorder

PONE-D-19-14998R1

Dear Dr. Horner,

We are pleased to inform you that your manuscript has been judged scientifically suitable for publication and will be formally accepted for publication once it complies with all outstanding technical requirements.

With kind regards,

Carla Treloar

Academic Editor

PLOS ONE

Additional Editor Comments (optional):

Reviewers' comments:

Reviewer's Responses to Questions

**Comments to the Author**

1. If the authors have adequately addressed your comments raised in a previous round of review and you feel that this manuscript is now acceptable for publication, you may indicate that here to bypass the “Comments to the Author” section, enter your conflict of interest statement in the “Confidential to Editor” section, and submit your "Accept" recommendation.

Reviewer #1: All comments have been addressed

Reviewer #2: All comments have been addressed

2. Is the manuscript technically sound, and do the data support the conclusions?

Reviewer #1: Yes

Reviewer #2: Yes

3. Has the statistical analysis been performed appropriately and rigorously? 

Reviewer #1: N/A

Reviewer #2: N/A

4. Have the authors made all data underlying the findings in their manuscript fully available?

Reviewer #1: Yes

Reviewer #2: Yes

5. Is the manuscript presented in an intelligible fashion and written in standard English?

Reviewer #1: Yes

Reviewer #2: Yes

6. Review Comments to the Author

Reviewer #1: I would like to commend the authors on the thorough revisions that they have made to this paper. The paper is an interesting qualitative analysis of the topic under investigation and the review of the relevant literature is more extensive.

Reviewer #2: (No Response)

7. PLOS authors have the option to publish the peer review history of their article (what does this mean?). If published, this will include your full peer review and any attached files.

Reviewer #1: No

Reviewer #2: No

---

## [Editor Report · Acceptance letter]

16 Oct 2019

PONE-D-19-14998R1 

“You’re kind of at war with yourself as a nurse”: Perspectives of Inpatient Nurses on Treating People who present with a Comorbid Opioid Use Disorder 

Dear Dr. Horner:

I am pleased to inform you that your manuscript has been deemed suitable for publication in PLOS ONE. Congratulations! Your manuscript is now with our production department. 

With kind regards,

on behalf of

Professor Carla Treloar 

Academic Editor

PLOS ONE